# Production of Recombinant Zika Virus Envelope Protein by Airlift Bioreactor as a New Subunit Vaccine Platform

**DOI:** 10.3390/ijms241813955

**Published:** 2023-09-11

**Authors:** Hernan H. M. da Costa, Monica Bielavsky, Diego J. B. Orts, Sergio Araujo, Patrícia P. Adriani, Juliana S. Nogueira, Renato M. Astray, Ramendra P. Pandey, Marcelo Lancellotti, Jair P. Cunha-Junior, Carlos R. Prudencio

**Affiliations:** 1Immunology Center, Adolfo Lutz Institute, São Paulo 01246-902, Brazil; 2Interunits Graduate Program in Biotechnology, University of São Paulo, São Paulo 05508-000, Brazil; 3Laboratory of Cardiobiology, Department of Biophysics, Paulista School of Medicine, Federal University of Sao Paulo, São Paulo 04023-062, Brazil; 4Skinzymes Biotechnology Ltd., São Paulo 05441-040, Brazil; 5Laboratory of Nanopharmaceuticals and Delivery Systems, Department of Pharmacology, Institute of Biomedical Sciences, University of São Paulo, São Paulo 05508-000, Brazil; 6Virology Center, Adolfo Lutz Institute, São Paulo 01246-902, Brazil; 7Multi-Purpose Laboratory Butantan Institute, São Paulo 05503-900, Brazil; 8School of Health Sciences and Technology, UPES University, Dehradun 248007, Uttarakhand, India; 9Faculty of Pharmaceutical Sciences—FCF, University of Campinas—UNICAMP, Campinas 13083-871, Brazil; 10Laboratory of Immunochemistry and Immunotechnology, Department of Immunology, Federal University of Uberlândia, Uberlândia 38405-317, Brazil

**Keywords:** envelope protein, recombinant protein, Zika Virus, rEZIKV, airlift bioreactor, FPLC

## Abstract

The Zika Virus (ZIKV) is an emerging arbovirus of great public health concern, particularly in the Americas after its last outbreak in 2015. There are still major challenges regarding disease control, and there is no ZIKV vaccine currently approved for human use. Among many different vaccine platforms currently under study, the recombinant envelope protein from Zika Virus (rEZIKV) constitutes an alternative option for vaccine development and has great potential for monitoring ZIKV infection and antibody response. This study describes a method to obtain a bioactive and functional rEZIKV using an *E. coli* expression system, with the aid of a 5-L airlift bioreactor and following an automated fast protein liquid chromatography (FPLC) protocol, capable of obtaining high yields of approximately 20 mg of recombinant protein per liter of bacterium cultures. The purified rEZIKV presented preserved antigenicity and immunogenicity. Our results show that the use of an airlift bioreactor for the production of rEZIKV is ideal for establishing protocols and further research on ZIKV vaccines bioprocess, representing a promising system for the production of a ZIKV envelope recombinant protein-based vaccine candidate.

## 1. Introduction

Zika Virus (ZIKV), an emerging arbovirus that can be transmitted by the biting of mosquitoes of the genus *Aedes*, was initially isolated in the Zika forest, in Uganda, in 1947 [1,2]. Even associated with some sporadic cases of human infection [3,4,5], ZIKV emerged as a great public health concern during the Americas’ disease outbreak in 2015 [6]: its vertical transmission was indeed correlated to newborns’ microcephaly, congenital malformations, and fetal death. Moreover, ZIKV infection is also associated with Guillain–Barré syndrome [7,8,9,10,11]. In addition, the presence of ZIKV in body fluids, such as semen, saliva, and urine, or blood transfusion and occupational exposure, represents possible sources for ZIKV transmission [12,13,14,15].

ZIKV is a member of the Flavivirus genus and its genome is a positive-sense single-stranded RNA (ssRNA) that encodes structural (S) and non-structural polyproteins (NS) [16,17,18]. The structural envelope protein (EZIKV) is responsible for mediating the binding and fusion of the virus to the host cell receptors and membrane, being one of the major determinants of ZIKV virulence and pathogenesis. EZIKV consists of three different domains: Domains I, II, and III. DI is located between DII and DIII, joining both. At the end of DII there is a fusion loop, which is responsible for interacting and fusing with host cells. Antibodies against DI and DII present weak neutralization effects [16,18,19], while DIII is the main target for viral neutralization [20]. These properties highlight the importance of the EZIKV protein in vaccine development due to its critical role in the stimulation of neutralizing antibodies, cellular immunity, and viral invasion [21].

Currently, no specific vaccines or drugs have been approved for ZIKV. Among many different vaccine pipelines under study, few studies are focused on recombinant subunits [22]. Recombinant technology is a feasible alternative for the development of ZIKV vaccines that has demonstrated immunogenicity and shows protective efficacy [23]. Moreover, the production of a recombinant Zika Virus envelope protein subunit (rEZIKV) can be a useful tool for monitoring immune responses in clinical trials in both mice and humans [24]. The development of efficient vaccines requires a combination of diverse strategies, and the production of protein subunits in bioreactors is a relatively easy procedure, allowing the synthesis of large amounts of a specific protein [25]. Recent progress in the area of recombinant technologies has paved the way to producing recombinant proteins that can be used as therapeutics, vaccines, and diagnostic reagents [26]. These platforms reduce costs and delivers reliable results while maintaining a degree of flexibility in the bioprocess. Only a few studies have evaluated bioprocesses for the production of ZIKV proteins, including the production of recombinant NS1 [25], wild-type ZIKV [27,28], and virus-like particles (VLPs) [29].

A wide variety of industrial and therapeutic proteins is produced by genetically modified *Escherichia coli*, which is easy to cultivate and also well characterized. However, there are only few studies regarding *E. coli* cultivation in the airlift bioreactor [30]. Stirred tank bioreactors are traditionally the most widely used equipment for culturing biological agents, as they provide the main requirements for the culture of microorganisms [31]. Although stirred tank bioreactors are considered to be the industrial standard for animal cell biotechnology, airlift bioreactors have been used in a number of large-scale processes [32]. Airlift bioreactors offer several advantages for large-scale bioprocesses, which are determined by the fluid dynamics and mass transfer characteristics [33]. The primary advantage of airlift bioreactors over other bioreactors is related to the homogeneous shear and stress forces distribution and the inexistence of focal points of energy dissipation throughout the reactor, which makes airlift bioreactors ideal for culturing shear sensitive cells, and also improves production on the bioprocess [33,34]. 

Here, we present a proof-of-concept regarding the production of a recombinant envelope protein of Zika Virus (rEZIKV) cloned in *E. coli* and cultured in a suitable 5 L (5-L) volume airlift bioreactor, and protein product purification by automated fast protein chromatography (FPLC). The purified recombinant protein presented preserved antigenicity and immunogenicity, demonstrated through serologic, in vivo evaluations, and immunomodulation properties.

## 2. Results

### 2.1. In Silico Analyses of Biochemical Properties of ZIKV Recombinant Envelope Protein (rEZIKV)

We performed an in silico three-dimensional visualization of the rEZIKV protein structure, which showed high similarity with the sequence deposited in the Protein Data Bank (PDB ID 5JHM). For this purpose, we have used the dimeric representation of rEZIKV, in which one monomer was represented as a cartoon diagram and the other as a surface epitope display, highlighting the envelope protein domains I, II, and III, and the fusion loop (Figure 1A).

The recombinant and the native EZIKV (PDB ID 5IRE) sequences were submitted to ProtParam software (https://web.expasy.org/protparam/ (accessed on 18 February 2023)) to perform biochemical analysis of the protein. The results show that rEZIKV has a molecular weight of approximately 43.6 kDa (291–690 amino acids), while the native EZIKV is approximately 54.4 kDa (291–794 amino acids). The most common amino acids in rEZIKV were Glycine (9.5%), Threonine (8.5%), and Valine (8.2%), and the less common were Tryptophan (1.8%), Phenylalanine (2.5%), and Tyrosine (2.5%); for native EZIKV, the most common were Glycine (10.9%), Leucine (8.9%), and Threonine (7.9%), and the less common were Tyrosine (2.0%), Tryptophan (2.0%), Glutamine (2.6%), and Cysteine (2.6%). The theoretical isoelectric point of rEZIKV was estimated as 5.93 and the extinction coefficient predicted as 53,400 M^−1^ cm^−1^. For native protein, the isoelectric point was predicted as 6.51 and the extinction coefficient as 69,900 M^−1^ cm^−1^. Moreover, the estimated half-life and stability for both proteins in *E. coli* cells was predicted to be higher than 10 h and stable, which led us to choose the *E. coli* to study the expansion of recombinant protein production (Figure 1B).

The NetNGlyc v1.0 and VaxiJen v2.0 softwares were used to analyze the rEZIKV in order to predict N-glycosylation sites and antigenicity. Structural analysis showed that both proteins present one N-glycosylation site, in amino acid N154. Prokaryotic systems, such as bacteria and archaea, generally lack the machinery for N-glycosylation, which is a type of glycosylation that occurs in eukaryotes. In addition, both proteins were predicted to be probable antigens (above the threshold value 0.4), having good antigenicity (VaxiJen score 0.6047 for rEZIKV and 0.6205 for native EZIKV) (Figure 1B).

### 2.2. Expression and Purification of rEZIKV

The airlift bioreactor batch was monitored until the end and parameters such as oxygenation, temperature, and pH variation were measured and adjusted, if necessary, to increase the yield of rEZIKV expression, providing a steady environment for bacterial growth. The parameters were evaluated hourly through the measurement of optical density at 600 nm and the dry weight (Figure 2). After 4.5 h of exposure to IPTG, the means of the parameters of the culture medium in the bioreactor monitored during the growth of transformed bacteria with the pET21a-rEZIKV were: temperature at 30.98 °C; pH of 7.22; and 35.86% of dissolved O_2_. These parameters indicate a stable environment for cell growth and rEZIKV expression, resulting in a dry weight increase of 9 mg (from 59 to 68 mg). 

After the expression phase, the bacterial suspension was subjected to lysis by liquid nitrogen and verified that a large amount of protein was aggregated into inclusion bodies, requiring an additional treatment with denaturing buffer. The protein was then purified by metal affinity chromatography using a HisTrap™ HP column in the fast protein liquid chromatography system ÄKTA™ Pure. Purified soluble proteins were analyzed by 12% SDS-PAGE under denaturing conditions (Figure 3A), presenting an approximately 43.6 kDa protein corresponding to the rEZIKV. All the fractions with peaks on the chromatogram were concentrated and used to estimate the yield. Analysis of purity using the ImageJ v1.53 software showed that the recombinant protein was obtained with approximately 80% of purity (Figure 3A), and the yield was estimated in 20 mg/L, quantified by spectrophotometry using the molar extinction coefficient obtained from ProtParam analysis (53,400 M^−1^ cm^−1^) on NanoDrop™ 2000.

The immunodetection test by Western blotting using a commercial anti-his-tag antibody presented a band of approximately 43.6 kDa, corresponding to the rEZIKV (Figure 3B). 

### 2.3. Humoral Immune Response Evaluation after rEZIKV Immunization in Mice 

After the expression and purification of rEZIKV, mice were immunized according to the schedule shown in Figure 4A. An ELISA was performed to evaluate the reactivity against the rEZIKV and mature ZIKV particles. When evaluating the post-third-dose serum from all immunized groups, the anti-OMV+ZIKV serum demonstrated high levels of reactivity against mature ZIKV viral particles, but did not react against the rEZIKV, both in ELISA and dot blot assays (Figure 4B,C). In contrast, the anti-rEZIKV serum showed a high reactivity against the rEZIKV and a weak reactivity against mature ZIKV (Figure 4B–D). The mock group serum showed no reactivity against any of the antigens.

The serum from the anti-rEZIKV group was also evaluated for the binding strength of IgG antibodies against the rEZIKV protein, measured as the avidity index (AI). The results showed that after the first dose, the rEZIKV immunization did not generate an immune response against the rEZIKV. However, after a boost dose, the reactivity against the rEZIKV had an approximately 17-fold increase in the ELISA index, maintaining increased levels after a third boost. This increase in immune responses also implied increased levels of avidity after the boost dose, reaching approximately 86% of avidity at this point, whereas after the last dose, the avidity level reached approximately 90% (Figure 4E). The mock group serum showed no reactivity against the rEZIKV after any of the doses. 

### 2.4. Cross-Reactivity Evaluation between rEZIKV and OMV+ZIKV Sera and DENV Serotypes

To evaluate the cross-reactivity of the rEZIKV and OMV+ZIKV immunization against the four DENV serotypes, an ELISA was performed using post-third-dose mice serum. Serum from animals immunized with rEZIKV did not react against DENV1 and DENV3 particles at any dilution. However, against DENV2 and DENV4 particles, it showed a slight reactivity (1.171 and 1.164 of Elisa Index, respectively), at 1:100 dilution. Serum from animals immunized with OMV+ZIKV reacted with all DENV serotypes in all dilutions tested. As expected, serum samples from the mock group did not react against any DENV serotype (Figure 5). 

### 2.5. rEZIKV Antigenic Potential Evaluation by ELISA

To evaluate the antigenic potential of rEZIKV, an ELISA was performed to detect specific anti-ZIKV IgG antibodies in human sera known to be negative for both DENV and ZIKV infection (n = 15); positive for DENV natural infection (n = 25); and positive for ZIKV natural infection (n = 25). The cut-off value was established based on the maximum sensitivity and specificity using a Two-Graph Receiver Operating Characteristic (TG-ROC) analysis. Thus, results from two different ELISA assays were normalized based on a single cut-off value. The use of rEZIKV as coating antigen on the ELISA assays detected specific IgG antibodies in the human sera of 19 patients who had recovered from ZIKV infection, resulting in a sensitivity of 76% (Figure 6A). In addition, the assay developed with rEZIKV showed a low cross-reactivity, being capable to differ DENV positive samples with a specificity of 96%, in which only one DENV positive serum sample reacted to rEZIKV (Figure 6A). The Receiver Operating Characteristic (ROC) curve corroborates the rEZIKV IgG-ELISA’s diagnostic performance for ZIKV or DENV positive samples compared to negative samples, showing a sensitivity of 76% and a specificity of 96% (Figure 6B,C).

### 2.6. mRNA Cytokine Expression Profile Evaluation by RT-qPCR

After BALB/c mice immunization, the ability of rEZIKV and OMV+ZIKV in inducing cellular immune response was investigated by measuring the mRNA expression level of cytokines after spleen cells in vitro antigen (rEZIKV or ZIKV particle) stimulation (Figure 7A). The results showed that splenocytes from animals immunized and stimulated with rEZIKV showed higher expression levels mainly for Interleukin (IL)-1β, IL-6, and Granulocyte-Macrophage Colony-Stimulating Factor (GM-CSF) cytokines (*p* < 0.05) (Figure 7B). The levels of the cytokine IL-4 (*p* = 0.0010) in the OMV+ZIKV immunized group that was further stimulated with rEZIKV were significantly higher than those in the mock group (Figure 7B). On the other hand, IL-4, IL-6, and GM-CSF (*p* < 0.05) showed a difference on spleen cells from mice immunized with OMV+ZIKV and further stimulated with ZIKV particles (Figure 7C). Lastly, the rEZIKV immunized group that was further stimulated with ZIKV particles showed an increase in mRNA levels for IL-1β, IL-2, IL-4, IL-6, IL-35, and Transforming Growth Factor (TGF)-β cytokines (*p* < 0.05) (Figure 7C).

## 3. Discussion

In this study, we present a reliable method for the production of a recombinant envelope protein of Zika Virus (rEZIKV) expressed in an *E. coli* system. To achieve a hight production, we used a 5-L airlift bioreactor and purified the recombinant protein product by an automated FPLC system. Many commercially available EZIKV are based on sequences from the African strain, but may not be easy to acquire or be affordable in ZIKV endemic areas in the Americas [35]. Additionally, the rEZIKV sequence used in this research was generated after alignment of 69 sequences from Brazilian isolates of ZIKV [36], which would be ideal for ELISA assays for serodiagnosis for ZIKV infection, mainly in the Americas, as these strains are of Asian–American lineage of ZIKV that currently caused outbreaks in these locations [35]. Bioinformatics showed no substantial differences, with the exception of the N154 glycosylation site inherent to the native form, and the rEZIKV was predicted to have good immunogenicity (Figure 1). A recent study using the VaxiJen tool reviewed eight works related to the development of potential vaccine candidates and concluded that the EZIKV protein was the primary target of vaccine design [37]. EZIKV glycosylation plays important roles in viral attachment and cell entry, replication, transmission, pathogenesis, and enhanced mosquito transmission, and/or increased vertebrate virulence of other flaviviruses, including West Nile virus, Japanese encephalitis virus, and tick-borne encephalitis virus [19,38,39,40]. Recently, no significant differences in immunogenicity nor in protection were observed after vaccination with recombinant EZIKV proteins expressed in a eukaryotic compared to a prokaryotic system [41]. In this study we pointed out that the presentation in dimeric form is a very relevant issue to be studied in future investigations since there are ways to modify such molecular features keeping quaternary epitopes [42,43]. This makes the production in *E. coli* a very attractive system to express rEZIKV.

Our results showed an expression and purification of approximately 20 mg/L of culture (Figure 2 and Figure 3), which was nearly 3 times higher than previously reported by Amaral et al. (2020) [36], and quite similar to those described by Liang et al. (2018) [41], who used an expression system based on *E. coli* and Drosophila cells. The 5-L airlift bioreactor used in this research allows automated control from the start to the end of the process, which improves tank circulation and oxygen transfer, as well as equalized shear forces in the reactor, with known, controlled, and reproducible cultivation conditions with greater homogeneity and productivity, contributing to a high yield and the expression of a high-quality protein [33,34,44].

There are many studies describing the expression and purification of recombinant EZIKV proteins using either *E. coli* [36,45,46], Drosophila cells [41], baculovirus [47], microalgae [48], plant [49], or mammalian cells such as HEK293T [24,35] in which the N-glycosylation on EZIKV was found to be an important factor for expression and secretion of this protein [50]. Regarding ZIKV recombinant protein production in a bioreactor, the ZIKV ΔNS1 protein was reported to be expressed using a bacterial system in a 6-L bioreactor, which reached a yield of 0.5 g per liter of culture [25]. Some comparisons allow us to understand the yield difference and guide us to improve our physicochemical parameters. Kanno and colleagues [25] upscaled the cultivation to a 6-L stirred tank bioreactor, using the conditions established by Response Surface Methodology (induction at 21 °C with 0.7 mM IPTG), improving the yield of ΔNS1. Moreover, one of the main reasons for the high production was possibly due to its small size (~17 kDa), compared to the rEZIKV (~43 kDa), which directly favors the inherent biochemical characteristics. To the best of our knowledge, there are no reports of yields for any recombinant envelope protein of ZIKV above 20 mg/L and there is no research focusing on rEZIKV production in an airlift bioreactor. 

In this scenario, the rEZIKV yield may be intrinsically related to its biochemical features. Thus, further improvements can be achieved by optimizing the bioprocess by the application of a specific statistical design to improve the yield and solubility of rEZIKV produced by *E. coli*. Our study is the first report on the production of rEZIKV using an airlift bioreactor, and the yield obtained so far is comparatively higher than levels obtained in other studies using traditional methods of expression of recombinant proteins. Therefore, our results give us great potential for improvement, and we aim to increase yields in future research.

To evaluate the immunogenicity of the recombinant protein, rEZIKV was applied in immunization trials. Following the last administration, we detected the presence of mouse anti-rEZIKV antibodies against the rEZIKV, confirming the immunogenicity of the recombinant protein (Figure 4). In contrast, the reactivity against the ZIKV mature particles was lower than the reactivity against the rEZIKV, which may indicate that some immunodominant epitopes of the native envelope protein of mature ZIKV were preserved in the rEZIKV (Figure 4B–D). The limited reaction of antibodies against mature viral particles may counts against the possible use of present antigen form. Interestingly, this same recombinant EZIKV was tested by another team who kindly provided the plasmid and showed that homologous EZIKV + poly (I:C) adjuvant prime-boost immunization was sufficient to induce robust EZIKV-specific humoral and cellular immune responses compared to other strategies that contemplate homologous DNA (pVAX-EZIKV) or heterologous (pVAX-EZIKV/EZIKV + poly (I:C), and vice-versa) candidates [36]. A central feature for rEZIKV vaccine development will be the choice of a proper adjuvant, as it can influence both specific humoral and cellular immune responses. Furthermore, tests were conducted to determine whether sera from mice immunized with OMV+ZIKV would react with the rEZIKV or ZIKV mature particles. An ELISA and immunoblots showed that anti-OMV+ZIKV antibodies displayed high reactivity against mature ZIKV particles. Similar results of reactivity were previously described by Martins et al. (2018) [51]. However, anti-OMV+ZIKV antibodies did not react with the rEZIKV (Figure 4B–D). These findings suggest that the recombinant protein subunit produced in bacterial cells may generate a different immune response to that generated by mature viral particles. 

Our immunization protocol showed an avidity antibody maturation (Figure 4E) and, interestingly, there was a possible association between antibody avidity with high neutralization antibody (nAb) titers for viral infections. Avidity indices below 40% are considered low, between 40% and 55% are considered to represent an “maturation zone”, and above 55% are considered high, in which there is a correlation between avidity and protection [52,53,54]. In our recent work [55], we demonstrated that DNA-based immunization in rabbits elicited high avidity maturation (approximately 92%), in correlation with a high nAb response both against live or pseudotyped SARS-CoV-2. This correlation can also be observed in natural infections, in which patients with severe cases of COVID-19 who survived the disease presented avidity indices higher than patients of milder cases or patients who died from COVID-19 [56]. It is fundamental to generate antibodies with high avidity and high titer of neutralizing antibodies. In addition, even testing only the recombinant form, we propose that the humoral immune response elicited by rEZIKV adds valuable information that may contribute to the immunization. High-avidity antibodies might increase immune protection, through improving antibody binding to virus particles and by stimulating effector functions as neutralization, opsonization, and complement activation for ZIKV infection [23,55,56]. 

We also evaluated the cross-reactivity between rEZIKV or OMV+ZIKV sera with DENV serotypes using an ELISA. The anti-rEZIKV antibodies did not react against DENV1 and DENV3 and presented a very low reaction against DENV2 and DENV4 antigens. On the contrary, anti-OMV+ZIKV antibodies reacted with the four DENV serotypes (Figure 5). Due to the high similarity between the envelope proteins of flaviviruses, mainly between ZIKV and the four DENV serotypes [13,40], the existence of common epitopes or homologous antigenic peptides accounted for the cross-reactivity between ZIKV and DENV [57]. The symmetric organization of all copies of envelope proteins on the virus creates new quaternary epitopes involving two or more molecules, found only on the surface of the virion. Previous data indicate that strongly neutralizing antibodies found in the sera of infected patients bound to epitopes that were preserved on the virion envelope but not on the recombinant E protein [58]. These dimer-dependent epitopes do not require high-order structural arrangements other than the dimeric conformation of the EZIKV, and appear to be highly conserved among flaviviruses. Antibodies against these epitopes were shown to have strong cross-reactivity against all DENV serotypes and ZIKV. Thus, we hypothesized that the immune response generated by OMV+ZIKV may be directed against a few dimer-dependent epitopes on the viral surface, which is highly conserved between ZIKV and the four DENV serotypes. The immune response generated against rEZIKV may be directed against epitopes that, naturally, are hidden under the viral surface, but in the recombinant conformation are more accessible to recognition by antibodies, which can lead to a response against epitopes less conserved among flaviviruses. To answer these hypotheses, further studies are needed, such as heterologous immunization and epitope mapping approaches [59,60]. 

Purified rEZIKV was used in an ELISA to detect ZIKV-specific IgG antibodies in human serum samples with previous exposure to ZIKV, DENV, or naïve healthy individuals. The ROC curves showed that the recombinant protein antigen can detect anti-ZIKV IgG in sera from ZIKV-infected patients and was useful to differentiate ZIKV-infected patient sera from both DENV patients and healthy individuals, with 76% sensitivity and 96% specificity (Figure 6). Although a sensitivity of 76.0% does not match clinical needs, in terms of diagnosis, the median sensitivity and specificity of different serological tests based on most recent published studies for ZIKV IgG ELISA are 92.9% (73.2–100%) for sensitivity and 93.8% (65.5–99.9%) for specificity [61]. Given the high specificity obtained, we believe that this test has the potential to be improved in the future either by better characterization of the cohorts or by technical improvements, since the balance of sensitivity and specificity is challenged by co-circulation of antigenically related viruses. In arbovirus tests, these intrinsic properties are inversely correlated. In addition, paired testing of acute phase and convalescence samples to show seroconversion compared to the standard Plaque Reduction Neutralization Tests (PRNT) is therefore a common procedure that can significantly increase the sensitivity to validate this assay. Envelope domain III has also been shown to be highly sensitive and specific for ZIKV diagnosis [62], while other domains, like NS1, have been used for ZIKV diagnosis [63,64] and some sensitive biosensors platforms have been developed also for ZIKV diagnosis [65]. 

While neutralizing antibodies are the primary correlates of protection against ZIKV, T cell immunity also plays an important role in controlling Zika infection [66]. Several experimental mouse models have explored the role of T cells in ZIKV protection [67,68,69]. Since cytokines play a key role in activating the immune system and generating an effective immune response, the identification of ZIKV-specific cytokines can help in the selection of target antigens and vaccine formulations. In our study, several transcripts to cytokines from spleen cells were evaluated (Figure 7). In these experiments, splenic cells from animals immunized with rEZIKV or OMV+ZIKV, after stimulation with rEZIKV or ZIKV mature particles, showed production of IL-1β, IL-6, GM-CSF, IL-2, IL-4, TGF-β, and IL-35 cytokine transcripts. Unexpectedly, none of the groups stimulated with different immunogens upregulate IFN-γ mRNA, a cytokine related to ZIKV protection in mice models. Thus, although immunization with rEZIKV induced a robust humoral response with the production of high-avidity antibodies, activation of Th1 cells was not achieved in our experimental conditions, suggesting that intrinsic molecular features from the rEZIKV protein may drive to non-polarized T cell responses. In addition, Amaral and colleagues elegantly demonstrated that splenocytes from mice immunized with homologous rEZIKV and poly (I:C) adjuvant presented a higher number of specific IFN-γ/TNF-α-producing T cells in comparison with heterologous prime-boost regimens, indicating that appropriated adjuvants may impact strongly in the type of T helper responses to rEZIKV [36]. Although cytokine data from mice cannot be directly transposed to humans, the evaluation of cytokines in patients with ZIKV infection also revealed a robust and multifunctional T cell response [70]. Several studies have explored the profile of cytokines in different clinical conditions related to Zika infections [70,71,72]. Therefore, the cytokine signature described so far for Zika disease needs to be studied more in the future.

In summary, our results show that the use of an airlift bioreactor for the production of rEZIKV is applicable. It can be used for establishing protocols and studies regarding the ZIKV vaccines bioprocess and represents a promising system for the production of ZIKV envelope recombinant protein, particularly in areas where ZIKV can cause outbreaks [73,74]. In addition, rEZIKV can be used to evaluate the seroconversion induced by vaccine candidates, since most vaccines under development use the envelope recombinant protein as the antigen of interest. 

## 4. Materials and Methods

### 4.1. Bacterial Cells, Plasmids, ZIKV, and DENV Inactivated Particles 

*Escherichia coli* cells from One Shot™ BL21 Star™ (DE3) pLysS kit (Invitrogen™, Carlsbad, CA, USA) compatible with pET vectors (DE3) were cultured in LB medium (1.0% peptone from casein, 0.5% yeast extract, and 1.0% NaCl). This strain was used for the expression of the recombinant protein. 

Professor Dr. Daniela Santoro Rosa from the Laboratory of Experimental Vaccines (LaVEx), Federal University of São Paulo (Unifesp), São Paulo, Brazil [36], kindly provided the plasmid expressing the recombinant envelope protein of Zika Virus (pET21a-rEZIKV).

Inactivated ZIKV and DENV particles were provided by Professor Dr. Marcelo Lancellotti (Biotechnology Laboratory, LABIOTEC, University of Campinas, São Paulo, Brazil). The protein concentration of inactivated virus samples was evaluated by spectrophotometry using the NanoDrop™ 2000 (Thermo Fisher Scientific™, Waltham, MA, USA).

### 4.2. Ethics Statements

Female BALB/c mice (6–8 weeks of age and approximately 25 g in weight) were acquired from the Animal House of the Interdisciplinary Procedures Center at Adolfo Lutz Institute. All mice were housed in a Ventilife Mice Mini-Isolator (Alesco^®^, Monte Mor, SP, Brazil) in a temperature-controlled light cycle facility in a Ventilife Ventilated Mouse Rack (Alesco^®^, Monte Mor, SP, Brazil), at 25 °C, and given antibiotic-free food and water ad libitum, in the experimental bioterium of the Immunology Center of Adolfo Lutz Institute. The Adolfo Lutz Institutional Committee for Animal Care and Use (CEUA/IAL no. 01/2022) approved all protocols, with all methods performed in accordance with relevant guidelines, regulations, and ethical principles in animal research adopted using the ARRIVE 2.0 guidelines [75].

The Adolfo Lutz Institutional Human Research Ethics Committee (CEP/IAL, CAAE no. 97331218.7.0000.0059) approved the use of human serum samples selected from IAL routine. The free and informed consent form was obtained from the research subjects and when it was impossible to obtain subjects’ signatures, the confidentiality term was used for the remaining samples sent to the laboratory for routine examinations.

### 4.3. Sequence and Structure Analysis

We employed the PyMol v2.5.4 software (DeLano Scientific LLC, San Carlos, CA, USA) to illustrate the structure of rEZIKV. For the rEZIKV structure, we used the available PDB file from the Protein Data Bank (code: 5JHM) [19]. For the structure and biochemical analysis, we used the ProtParam ExPASy Server [76]. In order to predict N-glycosylation sites, the protein sequence was submitted to NetNGlyc v1.0 software [77]. In addition, to predict protective antigens and subunit vaccines, the VaxiJen v2.0 server was used. Scores above the threshold value of 0.4 were considered a “probable antigen” [78].

### 4.4. Large-Scale Expression of ZIKV Recombinant Envelope Protein (rEZIKV) in an Airlift Bioreactor

For large-scale production of the recombinant protein, *E. Coli* cells were transformed with the PET21a-rEZIKV following an adapted protocol [79]. Fresh transformed bacterial cells were grown on 500 mL of LB liquid medium supplemented with ampicillin (100 µg/mL) at 37 °C overnight in a shaker at 250 rpm. Next morning, after the sterilization and calibration procedures, 4.5 L of sterile LB culture medium was kept in the bioreactor for 1 h until the stabilization of the following initial parameters: pH 7.4, temperature 37 °C, and 100% dissolved oxygen. The 500 mL of bacterial culture was then inoculated and grown for 1 h. At the end of the first hour, IPTG was added to the final concentration of 0.1 mM, the temperature was reduced to 31 °C, and the pH was maintained close to 7.0 with the use of 1 N NaOH solution, to optimize the expression of the recombinant protein. The protein expression was induced for 4.5 h, with samples taken hourly to measure optical density and dry weight. After induction, the culture was divided into 12 equal aliquots and harvested by centrifugation (5000× *g*, 1 h, 4 °C), the supernatant was discarded, and the pellet was resuspended in 5 mL of phosphate buffered saline solution (PBS) and stored at −80 °C. 

### 4.5. Recombinant Protein Purification

Bacterial cells were disrupted by freezing and thawing in liquid nitrogen followed by incubation with lysozyme, according to an adapted protocol [80]. The pellet fraction enriched with rEZIKV after lysis was resuspended in 10 mL of denaturing lysis buffer (50 mM NaH_2_PO_4_, 500 mM NaCl, 8 M Urea, and pH 8.0) and incubated overnight at 4 °C in an orbital homogenizer with constant stirring. The mixture was then centrifugated at 10,000× *g* for 2 h at 4 °C, and the supernatant was collected and proceeded to purification.

The supernatant from denaturing lysis was subjected to immobilized metal ion affinity chromatography (IMAC) on an automated fast protein liquid chromatography system (ÄKTA™ Pure, Cytiva Life Sciences^®^, Marlborough, MA, USA) using the 5 mL HisTrap™ High Performance (HP) column (GE Healthcare Life Sciences^®^, Chicago, IL, USA). The affinity protein purification was performed in three steps: Firstly, the supernatant was applied at a flow rate of 1 mL/min to the HisTrap™ HP column previously equilibrated with denaturing lysis buffer. The column was washed with three column volumes (CV) of the same buffer to remove the non-binding protein. The protein was then refolded with 10 CV of a linear gradient of 0 to 100% of renaturing buffer (50 mM NaH_2_PO_4_, 500 mM NaCl, 15% glycerol, 20 mM imidazole, pH 8.0, and 20 mM 2-mercaptoethanol), performed at a very slow flow rate of 0.5 mL/min. Lastly, the protein was eluted with 10 CV of a linear gradient of 0 to 100% elution buffer (50 mM NaH_2_PO_4_, 500 mM NaCl, 15% glycerol, 500 mM imidazole, pH 8.0, and 20 mM 2-mercaptoethanol). The his-tag was not removed from the rEZIKV protein after purification. The fractions containing the recombinant protein were selected and concentrated in Amicon^®^ Ultra-0.5 10 kDa (Merck Millipore, Burlington, MA, USA), then re-suspended in PBS buffer. The protein concentration was evaluated by spectrophotometry on NanoDrop™ 2000 (Thermo Fisher Scientific™, Waltham, MA, USA) using the molar extinction coefficient obtained from ProtParam analysis [81]. 

### 4.6. SDS-PAGE and Western Blot for Purity Analysis of rEZIKV

The purification fractions of rEZIKV were analyzed by SDS-PAGE under denaturing and reducing conditions. For this, samples were mixed with a sample buffer containing 10% 2-mercaptoethanol, incubated at 95 °C for 5 min and loaded in a 12% polyacrylamide gel. The run was performed at a voltage of 30 V overnight at 4 °C. The gel was fixed with a solution containing 10% methanol and 10% acetic acid (glacial) for 8 min, stained with a Coomassie G-250 Brilliant Blue dye solution for 15 min, and then de-stained with a solution containing 10% methanol and 10% acetic acid (glacial) at room temperature. To estimate the molecular weight of the proteins, the Amersham ECL Full-Range Rainbow Molecular Weight Marker (12–225 kDa) (Cytiva Life Sciences^®^, Marlborough, MA, USA) was used. ImageJ software [82,83] was used to estimate the percentage of rEZIKV in SDS-PAGE. The percentage was calculated considering the peak area correspondent to the rEZIKV band compared to the total area of all bands identified in the gel lane of purified protein.

For Western blot analysis, the 12% polyacrylamide gel was subsequently transferred to a 0.22 µm nitrocellulose membrane (Bio-Rad, Hercules, CA, USA) overnight at 4 °C using 50 V in a wet tank blotter system. For the immunodetection of the rEZIKV protein, a membrane was blocked with a 5% (*w*/*v*) solution of skim milk in PBS, at room temperature and under agitation, for 1 h. The membrane was washed 3 times with PBST 0.05% (PBS + 0.05% [*v*/*v*] Tween 20) for 3 min, each wash, under agitation. The membrane was incubated with a commercial rabbit anti-his-tag antibody (Sigma-Aldrich, St. Louis, MO, USA), at 1:1000 in a solution of 2.5% (*w*/*v*) skim milk in PBS, and stirred for 2 h at room temperature. The membrane was then washed 3 times with PBST 0.05% and incubated with goat anti-rabbit IgG conjugated to horseradish peroxidase (HRP) (Santa Cruz Biotechnology™, Dallas, TX, USA), at 1:10,000 in a solution of 2.5% (*w*/*v*) skim milk in PBS, and stirred for 1 h at room temperature. The membrane was washed 3 times with PBST 0.05% and incubated with SuperSignal™ West Pico PLUS Chemiluminescent Substrate kit solutions (Thermo Fisher Scientific™, Waltham, MA, USA) following the manufacturer’s recommendation. The reaction was visualized on the iBright™ CL 1500 imaging system (Thermo Fisher Scientific™, Waltham, MA, USA) and samples were compared to molecular weight standards used in SDS-PAGE.

### 4.7. BALB/c Mice Immunization

For immunization, 3 groups of 5 mice were used, and each group received 3 doses subcutaneously by injection in the back, with 10 µg of antigen at each dose diluted in sterile PBS with 50% (*v*/*v*) of complete (first dose) or incomplete (second and third doses) Freund’s adjuvant, at 21-day intervals (Figure 4A). Group 1 mice were immunized with the rEZIKV protein. Group 2 was immunized with outer membrane nanovesicles of *Neisseria meningitidis* conjugated to mature ZIKV (OMV+ZIKV), kindly provided by Dr. Marcelo Lancellotti [51]. Group 3 was the mock group and received only sterile PBS solution with 50% (*v*/*v*) of Freund’s adjuvant. Blood was collected via the submandibular vein on day 0 (pre-immune) and at a frequency of 21 days after each dose. In total, 4 blood collection time points were established: pre-immune, and 21 days after the first, second, and third doses. Mice were euthanized 3 weeks after the third dose by deep anesthesia with ketamine/xylazine use (140 mg/kg and 10 mg/kg, respectively). The blood samples were centrifuged at 3000× *g* for 15 min at room temperature and the serum was collected and transferred to new microtubes and stored at −20 °C. 

### 4.8. BALB/c Mice Splenocytes Stimulation in Cell Culture

After euthanasia, spleens from animals immunized with ZIKV antigens and mock group were harvested into complete RPMI-1640 Medium (10% heat-inactivated FBS, 0.075% sodium bicarbonate, 10 mM Hepes buffer, 100 U/mL penicillin G, 100 U/mL streptomycin sulfate, 1.5 mM L-glutamine, and 0.00035% 2-mercaptoethanol). The spleens were macerated and the cell suspension were collected. Red blood cells were eliminated by adding 5 mL of ice-cold endotoxin-free water followed by 1 wash with 10 mL of complete RPMI medium. Each cellular pellet was resuspended in RPMI medium, filtered through a sterile 70 µm nylon cell strainer to remove debris, and washed a final time in RPMI medium. Viable mononuclear cells were counted using Trypan blue dye exclusion in a Neubauer chamber and resuspended in complete RPMI medium. Splenocytes were seeded in 48-well (flat-bottom) plates at a concentration of 3 × 10^5^ cells per well (300 µL final volume) and stimulated with rEZIKV, ZIKV mature inactivated particles, and Concanavalin A (as positive control) at a concentration of 5 µg/mL, or only medium as a basal control, for 48 h.

### 4.9. mRNA Cytokine Expression from Stimulated Splenocytes

Total RNA was extracted from stimulated mouse splenocytes using TRIzol LS reagent (Life Technologies, Paisley, UK) and cDNA was synthesized using the GoScript Reverse Transcription System (Promega, Madison, WI, USA) according to the manufacturer’s instructions. Cellular expression of cytokines was analyzed by RT-qPCR using specific primers (Appendix A) [84,85,86,87] and GoTaq qPCR Master Mix kit (Promega, Madison, WI, USA); relative gene expression was normalized to endogenous expression of glyceraldehyde-3-phosphate dehydrogenase (GAPDH) mRNA per each representative basal control.

RT-qPCR was performed on an ABI 7500 Fast Real-Time PCR System (Applied Biosystems, Waltham, MA, USA) with the following conditions: 5 min at 95 °C initial denaturation; 40 × 15 s at 95 °C denaturation; 60 s at 60 °C primer annealing/elongation. The fluorescence was recorded during the annealing/elongation step in each cycle. We performed three technical replicates for each of the cytokines per stimulus to assess the relative quantification. The relative quantification (RQ) of a target gene in comparison with a reference gene was calculated according to the equation:RQ = Efficiency _target_ ^ΔCt target (basal−sample)^/Efficiency _reference_ ^ΔCt reference (basal−sample)^(1)

The mRNA expression was considered upregulated when fold change ≥ 2.0 and *p* < 0.05 [88].

### 4.10. Enzyme Immunoassay to Evaluate the Immunogenic Potential of rEZIKV

To assess the immunogenicity of rEZIKV, an ELISA was performed using pre- and post-immunization mouse sera. Briefly, two 96-well Nunc MaxiSorp™ flat-bottom ELISA plates (Thermo Fisher Scientific™, Waltham, MA, USA) were coated with 50 µL per well of a 5 µg/mL rEZIKV or mature ZIKV inactivated particles, diluted in PBS at 4 °C overnight. The next day, the plates were washed with PBST 0.01% (PBS + 0.01% [*v*/*v*] Tween 20) using the Washwell plate™ (Robonik, Thane, India), the wells were blocked with the addition of 100 µL of a 5% (*w*/*v*) solution of skim milk diluted in PBS, and the plates were incubated at 37 °C for 2 h. The wells were washed again with PBST 0.01% in the automatic washer. After washing, mouse sera were added to the wells, in duplicate, at a 1:50 dilution in a PBST 0.01% solution supplemented with 1% skim milk and the reaction was incubated at 37 °C for 1 h. The plates were washed again with PBST 0.01%, then 50 µL per well of goat anti-mouse IgG conjugated to HRP (Santa Cruz Biotechnology™, Dallas, TX, USA) was added at a dilution of 1:5000 in PBST 0.01% supplemented with 1% (*m*/*v*) of skim milk and incubated at 37 °C for 1 h. The plates were washed again with PBST 0.01%, then 50 µL per well of the One-Step solution—Linear TMB (Scienco Biotech, Santa Catarina, Brazil) was added and plates were incubated for 15 min at room temperature. The reaction was stopped with the addition of 50 µL per well of 1 N H_2_SO_4_. The plates were read at 450 nm using the MB-580 ELISA reader (Heales^®^, Shenzhen, China). The pre-immune serum of each group was used to calculate the cut-off of ELISA (mean plus two standard deviations), and the results were presented as an ELISA Index (Average Optical Density of sample divided by cut-off). 

### 4.11. Indirect ELISA for Evaluation of Anti-rEZIKV-Specific IgG Avidity Maturation 

Mice anti-rEZIKV IgG avidity maturation was evaluated by an ELISA using an adapted protocol previously described by our group [55,56]. Briefly, high-binding 96-well plates (Nunc MaxiSorp™ flat-bottom, Thermo Fisher Scientific™) were coated with 50 µL per well of 5 µg/mL of recombinant envelope protein diluted in PBS and incubated at 4 °C overnight. Plates were then washed 4 times with PBST 0.01% using an automated washer. After, 100 µL per well of 5% skim milk powder diluted in PBST 0.01% as a blocking solution was added to the plates and incubated for 1 h at 37 °C. After blocking, plates were washed with PBST 0.01% and incubated with 50 µL per well of mice-immunized serum samples, in duplicate, diluted 1:50 in PBST 0.01% supplemented with 1% skim milk for 1 h at 37 °C. Next, plates were washed with PBST 0.01%, then half of the plate was incubated in the presence of 1.5 M potassium thiocyanate (KSCN) and half of the plate in the presence of PBST 0.01% (50 µL/well), for 20 min at 37 °C. Plates were washed again and incubated with 50 µL of a 1:5000 dilution of goat anti-mouse IgG−horseradish peroxidase (HRP) conjugated antibody (Santa Cruz Biotechnology™, Dallas, TX, USA) diluted in PBST 0.01% supplemented with 1% skim milk for 1 h at 37 °C. Plates were washed again and incubated for 15 min with 50 µL/well of One-Step solution—Linear TMB (Scienco Biotech, Santa Catarina, Brazil). The reaction was stopped by the addition of 50 µL per well of 1 N sulfuric acid. The absorbance at 450 nm, using 630 nm as a reference, was measured using the MB-580 ELISA reader (Heales^®^, Shenzhen, China). The pre-immune serum absorbances were used to calculate the cut-off, using the mean plus two standard deviations. The avidity index was expressed as the absorbance mean value from the KSCN treated sample divided by the absorbance mean value from the nontreated sample and multiplied by 100%.

### 4.12. Detection of rEZIKV Protein by Immunoblotting

For Western blot assay, proteins were separated on a 12% polyacrylamide gel and subsequently transferred to a 0.22 µm nitrocellulose membrane (Bio-Rad, Hercules, CA, USA ), overnight at 4 °C and 50 V in a wet tank system. For dot blot assay, a 0.22 µm nitrocellulose membrane (Bio-Rad, Hercules, CA, USA) was coated with 5 µg of spot-selected rEZIKV and ZIKV mature inactivated particles, respectively, using the Bio-Dot Apparatus (Bio-Rad, Hercules, CA, USA ) and further divided into strips.

For the immunoblotting reaction, the membranes were blocked with a 5% (*w*/*v*) solution of skim milk in PBS, at room temperature and under agitation, for 1 h. The membranes were washed 3 times with PBST 0.05% (PBS + 0.05% [*v*/*v*] Tween 20) for 3 min, for each wash, under agitation. The membranes were incubated with the post-third-dose anti-rEZIKV, anti-OMV+ZIKV, or mock serum samples at a dilution of 1:5000 in a solution of 2.5% (*w*/*v*) skim milk in PBS and stirred for 1 h at room temperature. The membranes were washed 3 times with PBST 0.05%, then incubated with goat anti-mouse IgG conjugated to HRP (Santa Cruz Biotechnology™, Dallas, TX, USA) at 1:10,000 in a solution of 2.5% (*w*/*v*) skim milk in PBS and stirred for 1 h at room temperature. The membrane was washed 3 times with PBST 0.05% and incubated with SuperSignal™ West Pico PLUS Chemiluminescent Substrate kit solutions (Thermo Fisher Scientific™, Waltham, MA, USA) following the manufacturer’s recommendation. The reaction was visualized and images were recorded on the iBright™ CL 1500 imaging system (Thermo Fisher Scientific™, Waltham, MA, USA).

### 4.13. Cross-Reactivity Evaluation of rEZIKV Immunization by ELISA

To assess the cross-reactivity against the four DENV serotypes, an ELISA was performed using post-third-dose mice serum samples. Briefly, 96-well ELISA plates (Nunc MaxiSorp™ flat-bottom, Thermo Fisher Scientific™, Waltham, MA, USA) were incubated with 50 μL per well of a solution containing 10 µg/mL of each of the 4 inactivated DENV serotypes, diluted in PBS at 4 °C overnight. The plates were then washed with PBST 0.1% (PBS + 0.1% [*v*/*v*] Tween 20) using an automated washer and the wells blocked with the addition of 100 μL of 5% (*w*/*v*) skim milk solution diluted in PBS at 37 °C for 1 h. The wells were washed again with PBST 0.1%, and the plates were incubated with the mouse serum, in triplicate, at dilutions of 1:100, 1:200, 1:400, and 1:800 diluted in PBS solution supplemented with 5% (*w*/*v*) skim milk, and the reaction was incubated at 37 °C for 1 h. The plates were washed again with PBST 0.1% and incubated with 100 μL per well of goat anti-mouse IgG (whole molecule) conjugated to HRP (Sigma-Aldrich, St. Louis, MO, USA) at a dilution of 1:2500, in PBS solution supplemented with 5% (*w*/*v*) skim milk, at 37 °C for 1 h. The plates were washed again with PBST 0.1% and 100 μL per well of One-Step solution—Linear TMB (Scienco Biotech^®^, Santa Catarina, Brazil) was added and plates were incubated for 15 min at room temperature. The reaction was stopped with the addition of 50 µL per well of 1 N H_2_SO_4_. The plate was read at 450 nm using the MB-580 ELISA reader (Heales^®^, Shenzhen, China). The optical density of replicates of samples from the mock group, at each dilution, were used to calculate the cut-off (mean plus two standard deviations), and the results were presented as an ELISA index. 

### 4.14. Enzyme Immunoassay to Evaluate the Antigenic Potential of rEZIKV

To evaluate the antigenic potential of the rEZIKV, an ELISA was performed to detect specific anti-ZIKV IgG antibodies in the sera of 15 naïve healthy individuals for both DENV and ZIKV, 25 serum samples from patients with DENV natural infection confirmed by RT-qPCR, and 25 serum samples from patients with ZIKV natural infection confirmed by RT-qPCR. The serum samples were selected from the Virology Center of IAL routine. To ensure test reproducibility and concordance, two ELISA using the same protocol were performed on two different days. The average of all results from both days was used for plotting the graphs. Briefly, 96-well Nunc MaxiSorp™ flat-bottom ELISA plates (Thermo Fisher Scientific™, Waltham, MA, USA) were coated with 50 µL per well of a 5 µg/mL rEZIKV, diluted in PBS 1x at 4°C overnight. The next day, plates were washed with PBST 0.1% using the automated washer, the wells were blocked with the addition of 100 µL of PBST 0.1% solution containing 5% (*w*/*v*) skim milk, and the plates were incubated at 37 °C for 2 h. The wells were washed again with PBST 0.1% in the automatic washer. After washing, the human serum sample was added in a single well, at 1:100 diluted in a PBST 0.1% solution supplemented with 1% skim milk, and the reaction was incubated at 37 °C for 1 h. The plate was washed again with PBST 0.1%, then 50 µL per well of goat anti-human IgG (whole molecule) conjugated to HRP antibody (Sigma-Aldrich, St. Louis, MO, USA) was added at the dilution of 1:15,000 in a solution of PBST 0.1% supplemented with 1% (*m*/*v*) of skim milk, and the plate was incubated at 37 °C for 1 h. The plate was then washed again with PBST 0.1% and 50 µL per well of the One-Step solution—Linear TMB (Scienco Biotech^®^, Santa Catarina, Brazil) was added and incubated for 30 min at room temperature. The reaction was stopped with the addition of 50 µL per well of 1 N H_2_SO_4_. The plate was read at 450 nm with 630 nm as reference, using the MB-580 ELISA reader (Heales^®^, Shenzhen, China). The cut-off value was established based on the maximum sensitivity and specificity using a Two-Graph Receiver Operating Characteristic (TG-ROC) analysis [89] and the results were presented as an ELISA index.

### 4.15. Statistical Analyses

The data were evaluated for a normal distribution by Shapiro–Wilk normality tests. Statistical comparisons between ZIKV positive samples and DENV positive, or negative, samples were determined by one-way analysis of variance (ANOVA), with Tukey as post-test. Statistically significant differences of cytokine transcripts levels between immunized and mock groups in the RT-qPCR assay were determined by unpaired *t*-test or Mann–Whitney test, when appropriate. Statistically significant differences were indicated by *p* values < 0.05. Statistical analyses and graphics were performed using the GraphPad Prism v. 8.0.1 (GraphPad Software, San Diego, CA, USA).

## 5. Conclusions

In conclusion, this study demonstrates a successful production and purification of rEZIKV antigens in *E. coli* cells produced in an airlift bioreactor using standard conditions. The recombinant protein was able to raise an immune response in mice with high avidity. In addition, this study demonstrates that the rEZIKV antigen has the potential to be used in assays for seroprevalence, clinical differentiation between ZIKV and DENV, and monitoring vaccine clinical trials based on the ZIKV envelope recombinant protein-based vaccine candidate.

## Figures and Tables

**Figure 1 ijms-24-13955-f001:**
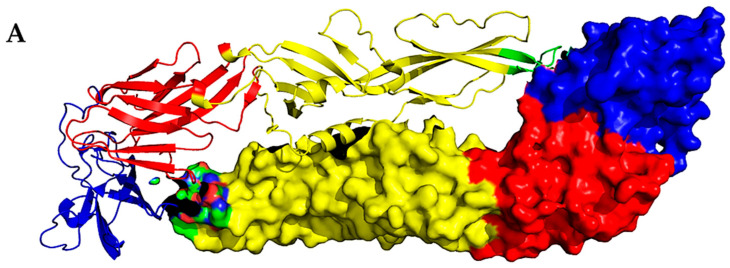
Bioinformatics analysis of biochemical properties of ZIKV envelope protein. (**A**) Dimer representation of EZIKV using a ribbon diagram and a surface epitope display representation (PDB ID 5JHM). EZIKV protein is represented as colored red (Domain I), yellow (Domain II), blue (Domain III), and green (fusion loop). (**B**) In silico analysis of recombinant EZIKV and native EZIKV biochemical properties.

**Figure 2 ijms-24-13955-f002:**
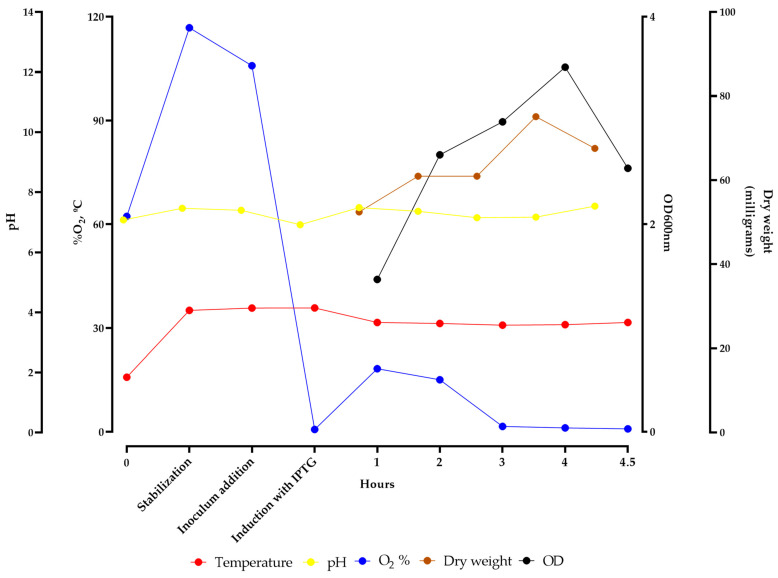
Parameters measured during expression of rEZIKV in airlift bioreactor. Measurement of temperature (red circles), pH variation (yellow circles), dissolved oxygen (blue circles), dry weight (brown circles), and optical density at 600 nm (black circles) during the time of bioreactor stabilization and bacteria growth pre- and post-induction with IPTG.

**Figure 3 ijms-24-13955-f003:**
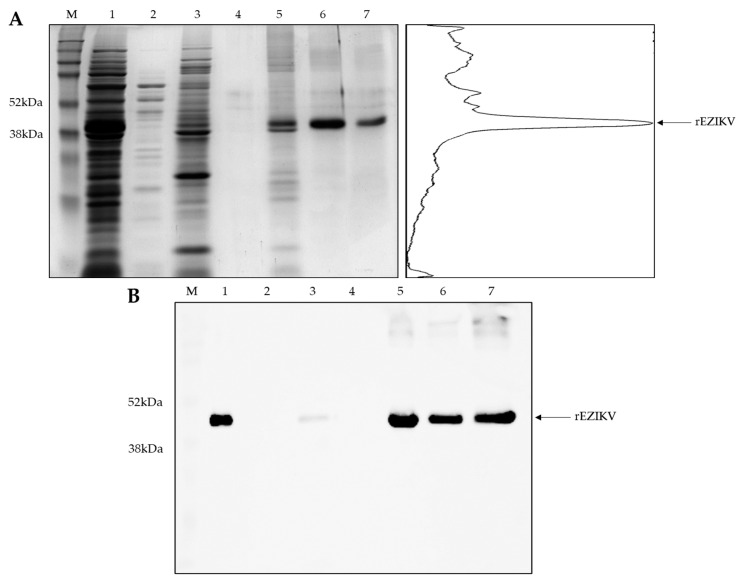
rEZIKV purification and immunoblotting. (**A**) SDS-PAGE in denaturing conditions showing the purification fractions and peak profile obtained by ImageJ software analysis using SDS-PAGE image from rEZIKV-purified fraction. (**B**) Western blot of rEZIKV purification steps using anti-his-tag. M: Amersham™ ECL™ Rainbow™ Marker; 1: Lysate post-denaturing treatment; 2: Flow-through; 3: Refolding process; 4: Wash step; 5: 1st eluate; 6: 2nd eluate; 7: 3rd eluate. The Western blot reaction was revealed using a chemiluminescent substrate, while the images were recorded using an imaging system and were adjusted for brightness and contrast.

**Figure 4 ijms-24-13955-f004:**
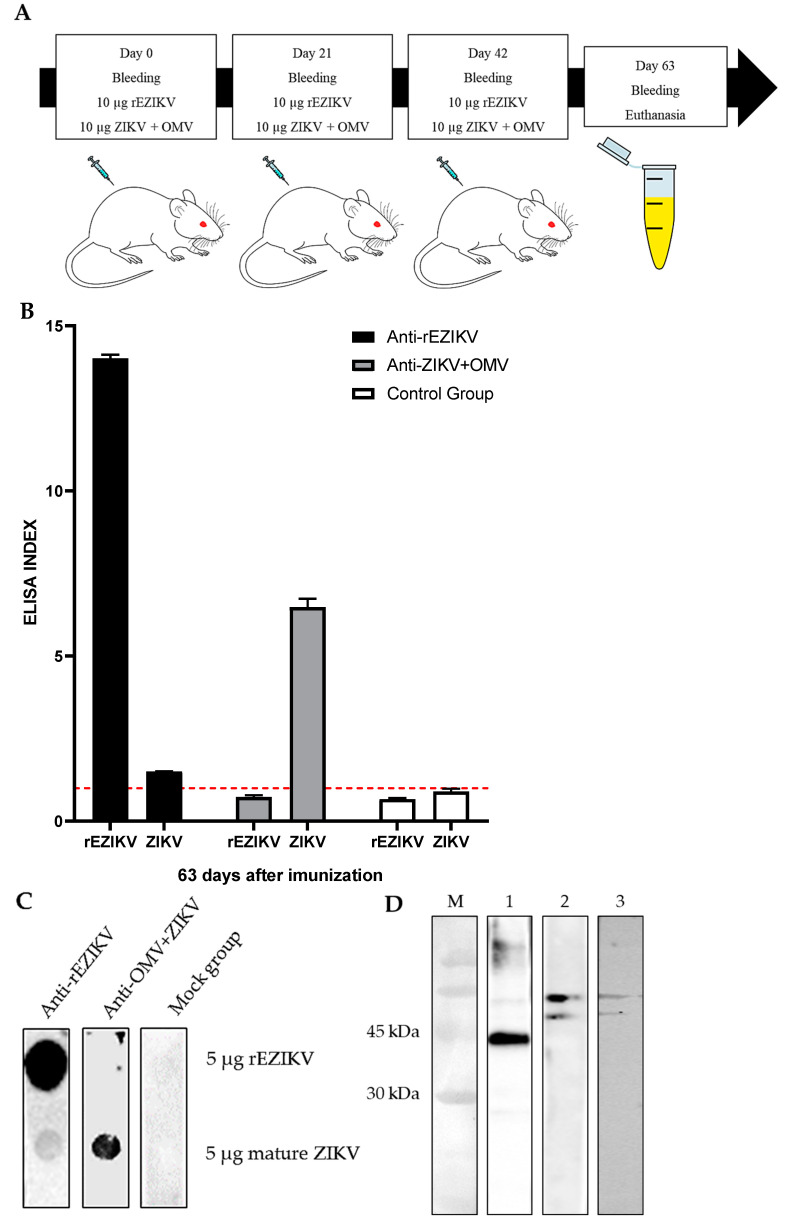
Evaluation of immune response after immunization using rEZIKV and OMV conjugated to ZIKV. (**A**) Immunization schedule. Mice received three doses at 21-day intervals and blood was collected via submandibular vein on day 0 and at a frequency of 21 days after each dose. Mice were euthanized 3 weeks after the last dose. (**B**) Levels of antibodies against rEZIKV and ZIKV mature particles as coating antigens, in mice anti-rEZIKV (black column), anti-OMV+ZIKV (grey column), and mock group (white column) post-third-dose serum. (**C**) Dot blot analysis using the anti-rEZIKV (left column), anti-OMV+ZIKV (middle column), and mock group (right column) post-third-dose serum, respectively. (**D**) Western blot analysis using post-third-dose mice serum. M: Amersham™ Low Molecular Weight Calibration Kit; 1: anti-rEZIKV serum; 2: anti-OMV+ZIKV serum; 3: mock serum. (**E**) Evaluation of anti-rEZIKV-specific IgG avidity against rEZIKV. Levels of anti-rEZIKV IgG (continuous black lines and dots), anti-rEZIKV IgG after KSCN treatment (discontinuous black lines and circles), and mock group IgG (continuous black line and triangle). The avidity index (AI %) is shown in discontinuous red lines and dots. The ELISA data were presented as an ELISA index (EI) and the dotted red lines show the cut-off value, in which EI > 1 results were considered positive. The blots reactions were revealed using a chemiluminescent substrate, while the images were recorded using an imaging system and were adjusted for brightness and contrast.

**Figure 5 ijms-24-13955-f005:**
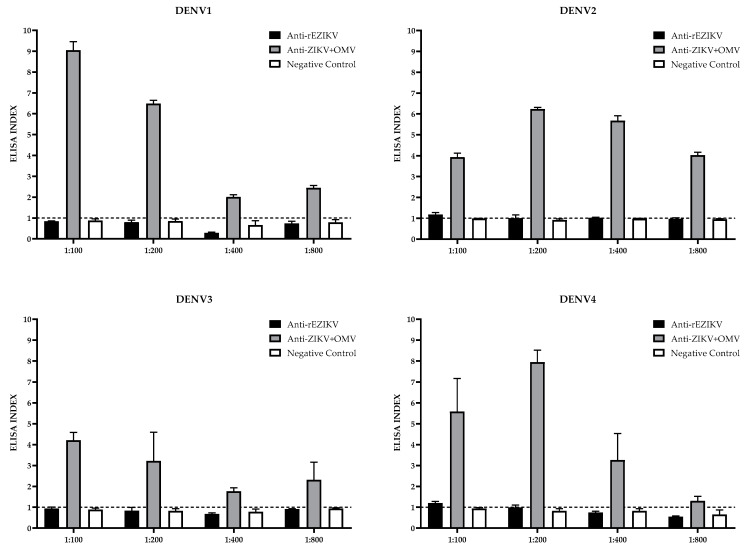
Evaluation of cross-reactivity against the four DENV serotypes by ELISA, using anti-rEZIKV (black column), anti-ZIKV (grey column), and mock group (white column) post-third-dose mice serum, respectively. Mock group serum was used to calculate the cut-off. The ELISA data were presented as an ELISA index (EI) and the dotted black line shows the cut-off value. EI > 1 was considered positive.

**Figure 6 ijms-24-13955-f006:**
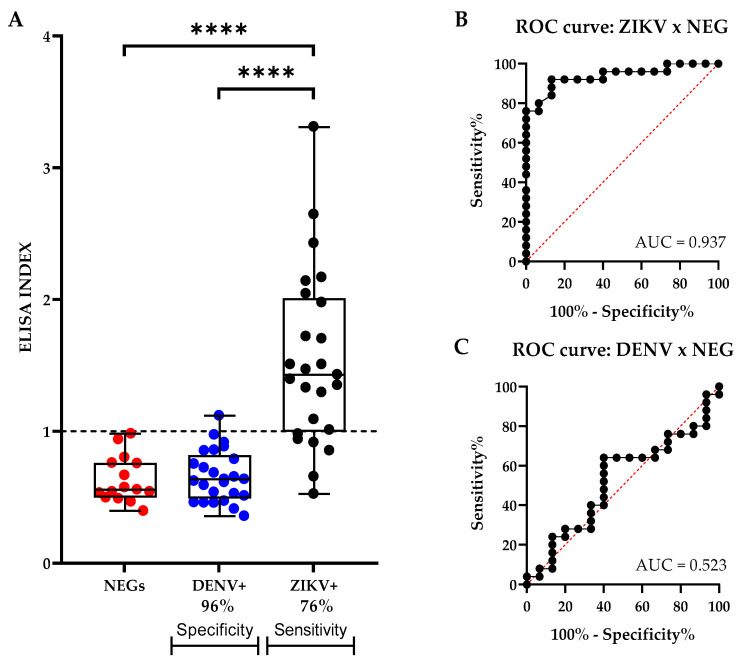
Antigenic potential of rEZIKV by ELISA. (**A**) Levels of antibodies against rEZIKV, in 15 negative serum samples for both DENV and ZIKV (red dots), 25 serum samples from patients with DENV natural infection (blue dots), and 25 serum samples from patients with ZIKV natural infections (black dots). (**B**) Receiver Operating Characteristic (ROC) curves show the rEZIKV IgG-ELISA’s diagnostic performance for ZIKV positive compared to negative samples. (**C**) Receiver Operating Characteristic (ROC) curves show the rEZIKV IgG-ELISA’s diagnostic performance for DENV positive samples compared to negative samples. The ELISA data were expressed as an ELISA index (EI) and the dotted black line shows the cut-off value. EI > 1 values were considered positive. Statistical comparisons between ZIKV positive samples and DENV positive, or negative, samples were determined by one-way analysis of variance (ANOVA), with Tukey as post-test. *p* values < 0.05 indicate statistically significant differences (**** *p* < 0.0001). AUC: Area under the curve.

**Figure 7 ijms-24-13955-f007:**
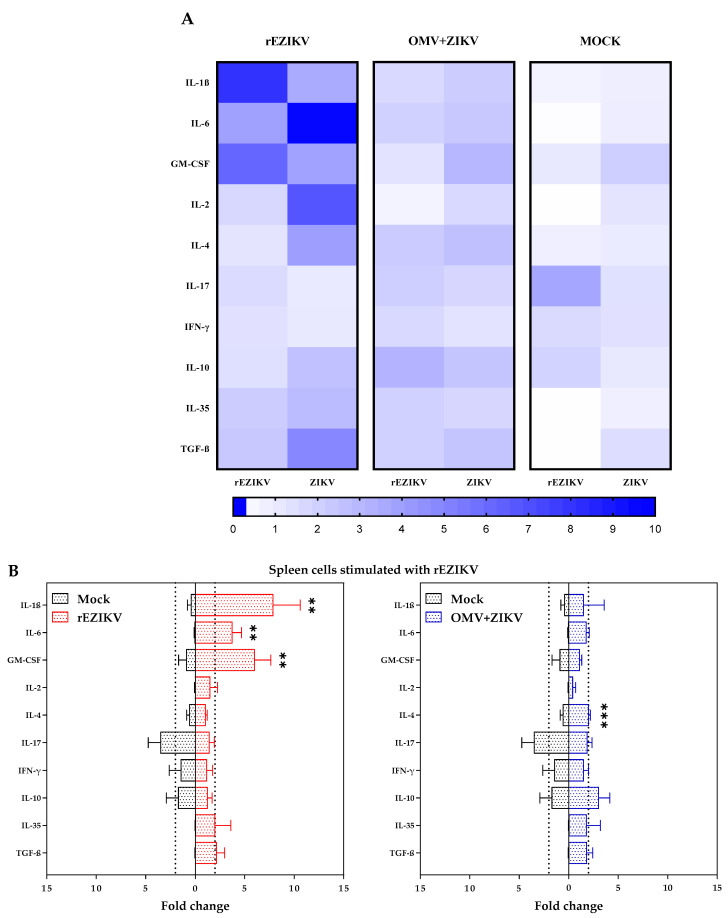
Cytokine mRNA profiles of BALB/c mice splenocytes cells stimulated with rEZIKV and ZIKV mature particles in vitro. (**A**) Heat map visualization comparing the mRNA cytokine expression profile between immunized and mock groups. (**B**) Comparison of mRNA transcript expression after rEZIKV stimulation in vitro. (**C**) Comparison of mRNA transcript expression after mature ZIKV antigens stimulation in vitro. mRNA expression of IL-1β, IL-6, GM-CSF, IL-2, IL-4, IL-17, IFN-γ, IL-10, IL-35, and TGF-β from splenocytes was detected by RT-qPCR. The mean basal mRNA expression for each group is equal to 1. The data are expressed as fold change of a target gene expression in comparison with a reference gene expression of GAPDH mRNA for each representative basal control. Red column: mice group immunized with rEZIKV. Blue column: mice group immunized with OMV+ZIKV. Black column: mock group. Statistically significant differences of cytokine transcripts levels between immunized and mock groups in RT-qPCR assay were determined by unpaired t-test or Mann–Whitney test, when appropriate. The data are expressed as the mean ± SD for each group. *p* values < 0.05 indicate statistically significant differences (* *p* < 0.05; ** *p* < 0.01; *** *p* < 0.0001).

## Data Availability

Correspondence and requests for materials should be addressed to C.R.P.

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
