# Peer review of "Production of Recombinant Zika Virus Envelope Protein by Airlift Bioreactor as a New Subunit Vaccine Platform"

_ijms, 2023, doi:10.3390/ijms241813955_

Round 1

Reviewer 1 Report

The present work by da Costa and colleagues describes the results obtained in a study on producing Zika Virus recombinant envelope protein (rEZIKV) with a bioreactor, in the view of vaccine production and infections monitoring. Zika virus (ZIKV) is a Flavivirus that captured scientific and public attention during the in 2015. Even other outbreaks were previously described, the Latin America one made evident the severe outcomes on newborns infected in-utero: many babies presented microcephaly and malformations, as well as fetal deaths occurred in significant percentage. Vaccination is therefore a key preventive tool, also considering the occasional and rapid ZIKV outbreak arising, as well as the potential extension of viral vectors habitat due to climate change.

Here follow specific observations.

TITLE

Clear and summarizing article content.

ABSTRACT

Clear and effective.

INTRODUCTION

LINE 43: avoid the Ae abbreviation, since it is not reported elsewhere in the text.

LINE 44-46: change this sentence, since 2015 is not that recent, as well as the concern was the outbreak itself. “Even associated with some sporadic cases of human infection [3–5], ZIKV emerged as great public health concern during the Americas disease outbreak in 2015 [6]: its vertical transmission was indeed correlated to newborns microcephaly, congenital malformations, and fetal death.”

LINE 51: put references at the end of the sentence [12-15].

LINE 52-53: change in “its genome is a positive-sense single-strand RNA”.

LINE 57-61: please rephrase, trying to avoid repetitions (i.e., neutralizing antibodies).

LINE 73: change in “process, capable…”

RESULTS

LINE 102: what is PBD? Only the acronym appears in the text: please provide full name.

LIEN 106-114: is it possible to compare recombinant and native proteins characteristics, as in figure 1B, to underline presence or absence of relevant differences? Most common amino acids are reported, but having some additional information of rEZIKV vs ZIKV would help in understand and contextualize results: for example, the comparison could be used to discuss the recombinant immunogenicity.

LINE 115-119: is there any information about glycosylation of native protein? Is the N154 glycosylation the only difference between native and recombinant proteins? Could this impact immune response, nAb production and avidity? Please report a range or another means of interpretation for the VaxiJen score.

DISCUSSION

LINE 296-318: is it possible to add some information on expected/optimal yield of antigens production by airlift bioreactor? In the study, the rEZIKV amount was 20 mg/L, 3-fold more than E.coli and similar to Drosophila, but much lower than bioreactor (0.5 g/L).

LINE 319-332: this point is major weakness of rEZIKV. The limited reaction of Abs against mature viral particles counts against the possible use of present antigen form in vaccine formulation.

LINE 342-345: avidity is linked to antibodies present in the serum; for this reason, even an increase in avidity was found in the present work, the data has limited usefulness, considering what reported in the previous paragraph.

LINE 355-359: the limited cross-reactivity could be due also to different glycosylation between recombinant and native antigen, emphasizing the limited usefulness of this Ag form for immunization.

LINE 372: a sensitivity of 76.0% does not match clinical needs, in terms of diagnosis.

LINE 381-418: what is the relevance of cellular response in ZIKV infection? In other words, are there any data on the expression of the reported molecules in ZIKV patients? Studying cytokine and other molecules expression is for sure interesting, but it should also be inserted in a clinical context.

LINE 423-426: given the limited sensitivity, this conclusion is quite overhasty.

Reviewer 2 Report

da Costa HHM present a novel use of air-lift bioreactors for the enhanced production of ZIKA envelope recombinant protein for use in vaccination. 

Major concerns:

1. E. coli don't usually glycosylate protein. There's some very niche lab strains that do. Methods say you used commercial BL21 pLysS making Figure 1 a supplementary figure at best. Prabhu SK et al. (2021) Bioorg Med Chem, Ding N et al. (2017) Biochem Phys Res Commun. If glycosylation is the focus of this study it should be scientifically confirmed on the platform. 

2. COI of P Adriani not listed in COIs

3. There is no challenge data. Ovalbumin elicits an immune response but doesn't make it effective against infection. This would have to be repeated in ZIKA-infectable mice (e.g. Ifnar1-/- mice).

Round 2

Reviewer 1 Report

Thank you to the authors for the great effort and clarification, that significantly improved the work, making it more clear and complete.

Reviewer 2 Report

Comments supplied are sufficient, but a challenge study is highly recommended in the future.

Minor point: Title should be subunit protein not vaccine.